# OpenReview forum: "Stronger Universal and Transfer Attacks by Suppressing Refusals"
_NeurIPS.cc/2024/Workshop/SafeGenAi — SafeGenAi Poster_

### Official Review · Reviewer_TsbV · 2024-10-08
**Interesting improvement upon GCG**

**Rating:** 6
**Confidence:** 4

**Review:**

The paper presents an improvement in the optimization method for GCG that directly tries to optimize for refusal suppression.

Strengths:
* The paper presents a nice improvement on GCG that increases transferability.
* I think the main contributions of the paper should be the findings and implications of including new interpretability-derived terms in the GCG loss and how this improves transferability. As I explain later, I do not fully understand the current section 3.

Weaknesses:
* It is mostly a straight-forward combination of existing work (GCG + refusal supression). Yet, I think contribution is relevant to the venue.
* The authors claims in Section 3 that
> selecting suffixes optimized for individual behaviors is not necessarily optimal and that universal suffixes are more generalizable across models leading to higher transferability.

If I recall correctly, this phenomenon is also explored in detail in the original GCG paper where the best performing suffixes where optimized on several prompts and models simultaneously. I think this needs a major modification for camera ready. The paper spends quite some time discussing the effectiveness of these "universal suffixes" that I did not completely understand how they differ from the original GCG methods.

* I think the writing of the paper can generally be improved. It is sometimes hard to understand what are the methods, main contributions and differences with previous work.
* The authors should report the performance of baseline GCG optimized on LLaMA-3 on frontier models. The current table 3 does not allow for this comparison since the best transferrable suffix is found on LLaMA-3 but the baseline is only computed on LLaMA-2.

Suggestions:
* I encourage the authors to optimize prefixes on several models simultaneously since finding suffixes that can suppress several refusal directions simultaneously may improve transferability.

---

### Official Review · Reviewer_MQFw · 2024-10-09

**Rating:** 4
**Confidence:** 5

**Review:**

Summary:

The paper deals with adversarial suffix-based jailbreak attacks. The authors observe suffixes from GCG are inherently universal and transferable and make the hypothesis that these suffixes deactivate the “safety feature”. To further leverage this existing interpretability, they introduce a new loss term in the conventional GCG optimization. They achieve 56% ASR on Llama-3 and transfer to close-source models with high ASR. They also achieve 96% ASR against Representation Rerouting defense.

Pros:
- Red-teaming efforts are important to identify the risks and build safer models.
- The exploration of neuron-level interpretation and its application in jailbreak attacks seems to be interesting, especially for open-source models.

Cons:
- The novelty is incremental. The introduction of “safety feature” has been explored in previous work [1] and the additional loss term inspired by it seems to be incremental.
- The evaluations are unclear and weak. What dataset and experimental setting are used in Table 1? I guess it is AdvBench by default. Then the evaluation on only 50 random prompts is not enough to claim the suffix to be universal and transferable attack because there are many redundant prompts in AdvBench. It would be more convincing if evaluations were conducted on HarmBench[2] and JailbreakBench [3].
- The claim “surprising generalization achieved from single-behavior optimization” in Table 1 is not accurate. Selecting the best suffix among 50 different behaviors with the best performance on the ‘test set’ is not reasonable for claiming inherently universal and transferable. This conclusion could be totally different when testing on another held-out set.  GCG is optimized on not only a single behavior but also multiple behaviors with gradients from multiple models. The transferability is verified on a held-out set with a size of 388. The author should check the details carefully before making their conclusion.
- Even if the authors explore the refusal vector and its application in GCG optimization, how does this connect to the observation in Sec. 3? How does it verify “We believe this suggests some adversarial suffixes operate by directly deactivating the safety feature rather than simply forcing it to repeat a specific target string”?
- The writing needs to be improved and clear notations should be provided in Sec.4. It is hard to follow and understand.

Reference:

[1] Arditi, A., Obeso, O., Syed, A., Paleka, D., Rimsky, N., Gurnee, W. and Nanda, N., 2024. Refusal in Language Models Is Mediated by a Single Direction. arXiv.

[2] Mazeika, M., Phan, L., Yin, X., Zou, A., Wang, Z., Mu, N., Sakhaee, E., Li, N., Basart, S., Li, B. and Forsyth, D., 2024. Harmbench: A standardized evaluation framework for automated red teaming and robust refusal. ICML.

[3] Chao, P., Debenedetti, E., Robey, A., Andriushchenko, M., Croce, F., Sehwag, V., Dobriban, E., Flammarion, N., Pappas, G.J., Tramer, F. and Hassani, H., 2024. Jailbreakbench: An open robustness benchmark for jailbreaking large language models. NeurIPS.

---

### Official Review · Reviewer_LDVB · 2024-10-10
**Although the method modifications are marginal, the contribution of this work remains meaningful, considering the generalization capability tests of GCG and incorporating the model’s internal safety mechanisms into an automated jailbreaking algorithm.**

**Rating:** 6
**Confidence:** 5

**Review:**

This paper extends the Greedy Coordinate Gradient (GCG) approach by exploring two key enhancements. First, the adversarial suffix discovered by GCG for a single model or request can be transferred to attack other models, showcasing its potential for transferability. Second, a new loss term is introduced into the GCG framework to improve its overall performance in adversarial settings.

Although the modifications to the method are marginal, the contribution of this work remains meaningful. On one hand, the study tests the generalization capability of GCG across different models, while on the other, it aims to enhance jailbreak performance by incorporating the model’s internal safety mechanisms into an automated jailbreaking algorithm.

However, my main concern lies in the fact that the experiments were primarily conducted on classical LLMs. Given the rapid updates and advancements in commercial LLMs, there is a risk that the findings could become outdated for more recent LLM versions. The fast-paced development of LLMs raises questions about the continued relevance of the results as models evolve. Besides, the attack setting in this paper is so white-box, however, accessing internal information of commercial LLMs is hard work.